# Hepatic *IFNL4* Gene Activation in Hepatocellular Carcinoma Patients with Regard to Etiology

**DOI:** 10.3390/ijms22157803

**Published:** 2021-07-21

**Authors:** Henriette Huschka, Sabine Mihm

**Affiliations:** Department of Gastroenterology, Gastrointestinal Oncology and Endocrinology, University Medical Center, 37075 Goettingen, Germany; Henriette.huschka@stud.uni-goettingen.de

**Keywords:** hepatitis C virus (HCV), hepatocellular carcinoma (HCC), interferon-stimulated gene (ISG), interferon lambda 4 (*IFNL4*), rs368234815, rs11322783, The Cancer Genome Atlas (TCGA)

## Abstract

Hepatocellular carcinoma (HCC) is a malignancy with a leading lethality. The etiology is quite diverse, ranging from viral infections to metabolic disorders or intoxications, and associates with specific somatic mutational patterns and specific host immunological phenotypes. Particularly, hepatitis C virus (HCV)-infected liver is featured by an activation of interferon (IFN)-stimulated genes (ISGs; IFN signature), which we suppose is driven by type III *IFNL4*. Taking advantage of the TCGA collection of HCC patients of various different etiologies, this study aimed at validating our previous findings on hepatic *IFNL4* gene activation in HCV infection in an independent and larger cohort of patients with advanced liver disease. In a cohort of *n* = 377 cases, the entirety of the sequencing data was used to assess the *IFNL* genotypes, and the cases were stratified for etiology. The number of *IFNL4* transcripts within nonmalignant and malignant tissues was found to be more abundant in patients with HCV or HCV/HBV infections when compared to other risk factors. Moreover, in patients with HCV infection as a risk factor, a close, positive relationship was found between ISG activation and the number of functional *IFNL4* transcripts. Data on this independent TCGA sample support the concept of an *IFNL4*-dependent HCV-driven activation of hepatic ISGs. In addition to that, they add to the understanding of etiology-related host immunological phenotypes in HCC.

## 1. Introduction

Hepatocellular carcinoma (HCC) is a leading cause of cancer-related deaths [1], with a 5-year survival rate < 20% [2,3]. The etiology is quite diverse, ranging from chronic infections with hepatitis B virus (HBV) or hepatitis C virus (HCV) to metabolic liver diseases like non-alcoholic fatty liver disease (NAFLD) or non-alcoholic steatohepatitis (NASH), alcohol-related liver disease (ALD), exposure to Aflatoxin B1, or less frequent genetic disorders such as hereditary hemochromatosis [4,5]. These risk factors were shown to associate with mutational patterns in HCC, e.g., inactivating mutations within the tumor-suppressor *TP53* gene in HBV-related HCCs or *TERT*-promoter mutations activating telomerase expression in ALD-related liver cancer [5,6,7]. In contrast, no relationship has been seen so far for HCV-related HCCs. These were found, however, to be associated with the development and the shape of the host-adaptive antitumor immune response towards tumor-associated antigens (TAAs) [8] and, recently, with specific phenotypic immunological profiles of CD8^+^ T cells affecting antitumor immune surveillance [9].

Beyond etiology-related somatic mutational landscapes and etiology-related acquired immunological phenotypes in HCC, particularly, HCV-infected (nonmalignant) livers are known to be featured by the activation of interferon (IFN)-stimulated genes (ISGs), which are mediators of the innate immune response. This IFN signature is featured by myxovirus resistance protein A (*Mx1*), IFN-induced protein 44 (*IFI44*), and IFN-induced protein with tetratricopeptide repeats 1 (*IFIT1*) and differs from gene expression profiles in nonviral or metabolic liver disorders and, notably, in HBV infections in man [10,11,12]. Transcripts of IFNs themselves, in contrast, i.e., of type I IFNs (IFN-α_n_/β) or type III IFNs (IFN-λ_1–3_), were hardly found to be activated in human hepatitis C liver tissue samples [10,12,13]. *IFNL4* was the latest discovered type III IFN [14]. Different to other type I or type III IFN genes, *IFNL4* harbors a gain- or loss-of-function dinucleotide polymorphism: the phylogenetically older allele ΔG translates into a functional IFN-λ_4_ protein, while the variant allele TT causes a frameshift disrupting the *IFNL4* open reading frame (ORF), thereby abrogating its translation [14]. This germline polymorphism is driving the pseudogenization of the *IFNL4* gene and, thus, predisposes a subpopulation of human beings only to express a functional IFN-λ_4_ protein [15]. By quantifying the *IFNL4* transcripts in liver biopsy specimens from a broad panel of various different nonmalignant liver disorders, we found *IFNL4* transcripts (i) detectable exclusively in hepatitis C patients, (ii) their total number to be correlated to hepatic viral load, and (iii) the number of functional but not that of nonfunctional transcripts to be associated with the activation of ISGs [16]. These data, together with affirming findings by others [17,18], are suggestive for HCV being an activator of *IFNL4* transcription in human liver, fostering an IFN signature in those individuals who carry the functional *IFNL4* variant. Yet, they demand validation.

The Cancer Genome Atlas (TCGA) database provides comprehensive demographic, clinical, and genetic datasets of >11,000 cancer patients across 33 entities, among them 377 from patients with HCC. These cases are of a wide variety of etiologies, including HBV infection, HCV infection, HBV/HCV coinfections, or of nonviral causes. This study aimed at finding clinical evidence for or against hepatic *IFNL4* gene activation in HCV infection in an independent and larger cohort of patients with advanced, i.e., malignant liver disease from TCGA.

## 2. Results

### 2.1. The TCGA HCC Patient Sample with Regard to Etiology

The TCGA LIHC dataset contains 377 cases of liver cancer diagnosed as HCC (C22.0). It encompasses demographical and clinical documentation, transcriptome profiling, and whole-exome sequencing data of tumorous and nontumorous tissue samples. With a mean age of 59.3 y at diagnosis and a male:female ratio of 2.1, the majority of patients are of Caucasian (*n* = 187) or Asian (*n* = 161) ethnicities, with 17 patients of African and 2 of indigenous ancestry (Table 1).

The documented HCC risk factors included ethanol consumption, NAFLD, hemochromatosis, HBV infection, HCV infection, tobacco use, insulin-dependent diabetes mellitus (IDDM), or combinations of those. Based on these records, we allocated patients into groups with no history of primary risk factors, however, unavailable hepatitis serology (*n* = 54), with nonviral risk factors (*n* = 43) and unavailable serology alike, with serologically confirmed previous or active HBV or HCV infection (*n* = 144 or *n* = 48, respectively), or with serologically confirmed previous or active HBV/HCV coinfections (*n* = 88) (Table 1). Expectedly, a history of HBV infection or HBV/HCV coinfections were found to be more prevalent among patients of Asian ancestry, while isolated HCV infections and etiologies of unknown or nonviral risks, on the contrary, predominated among patients of Caucasian ancestry. Patients with HBV-related HCCs presented more advanced tumor grades, while patients with HBV/HCV coinfections featured higher tumor stages (Table 1).

### 2.2. IFN Gene Signature in Nonmalignant and in Malignant Liver Tissues

Transcriptome profiles are available from 371 tumors and 50 corresponding nontumorous tissue specimens. This enabled the analysis of type I and type III IFN and IFN effector gene activation in nonmalignant liver tissue with regard to primary diseases and, also, in corresponding malignant tissue. Type I IFN gene transcription was representatively read by the amount of mRNAs of three single species: *IFNA2*, *IFNA8*, and *IFNA21* and by *IFNB*, while all four type III IFN genes, *IFNL1-4*, were regarded. The transcriptional activation of type II *IFNG*, albumin, and GAPDH as references was considered as well. Regarding nonmalignant liver tissue, type I IFNs—particularly, IFN-α mRNAs—were undetectable in almost all samples: in patients with nonviral etiology and, also, in those with HBV or HCV infections (Figure 1). Among type III IFN mRNAs, *IFNL1* was found to be a more abundant species, but all of them, including *IFNL4*, were detectable preferentially in patients with HCV infections. This is in line with previous findings [10,12,13].

By analyzing the corresponding malignant samples, IFN gene expression was rare, and it was unrelated to their matched counterparts. Of note, the tumor tissue was found to feature significantly less albumin transcripts, which might reflect dedifferentiation (Figure 1). Moreover, the malignant samples had significant higher levels of GAPDH mRNA, which might reflect an anaerobic metabolism (Figure 1).

By analyzing the whole set of tumor tissue samples, likewise, detectable levels of *IFNL4* transcripts were present in a small portion of the samples only (56/371) (Figure 2A). They were found to be more abundant in patients with HCV-related HCCs than in those without a history of HCV infection (Figure 2B). Being, for the most part, on an overall low level, the mRNA levels beyond that (>1000 FPKM-UQ) were noticeable, with one exception for HCC patients with HCV or HCV/HBV coinfection histories only (1/236 vs. 7/135, χ^2^-test *p* = 0.0024) (Figure 2A). 

### 2.3. ISG Gene Activation with Regard to IFNL Genotypes

Different to its transcription, the translation of the *IFNL4* gene into a functional IFN-λ_4_ protein requires the *IFNL4* rs368234815 ΔG variant. TCGA whole-exome and RNA sequencing data were surveyed by aligning the genomic reads to a reference genome. As coverage was found to be low at *IFNL4* rs368234815, proxies were identified by employing the LDproxy application of the Web-based tool LDlink. Within the *IFNL3/4* region, these were *IFNL4* rs117648444, *IFNL3/4* rs12979860, and *IFNL3* rs4803217 (Table 2). The latter one had a sufficient high coverage (>30) to allow genotyping. Moreover, the allele frequencies were rather close to each other, not only globally but also when the Asian and European populations were considered separately (D’ = 0.965 and r^2^ = 0.885 and D’= 0.991 and r^2^ = 0.968 for the Asian (EAS and SAS) and European (EUR) populations, respectively). *IFNL3* rs4803217 genotypes were thus taken as a surrogate for *IFNL4* rs368234815.

The call rate of the *IFNL3* rs4803217 genotypes was 99.7% (376/377). For the entire cohort, the genotype distribution was found to amount up to 227:123:26 for CC:CA:AA (Table 1), matching the Hardy–Weinberg equilibrium (HWE) (*p* = 0.7673).

Similar to hepatic ISG expression, which is known to be stimulated in HCV-infections in nontumorous conditions, hepatic ISG expression was found to be activated in malignant tissue—specifically in patients with HCV infections as a risk factor—as well (*p* = 0.0027, *p* = 0.0001, and *p* = 0.0148 for *Mx1*, *IFI44*, and *IFIT1*, respectively, Kruskal–Wallis test, Appendix A Appendix A). By combining all available data from the nonmalignant and malignant samples, a positive correlation for ISG mRNA expression and the number of functional *IFNL4* transcripts could be found in the group of patients with HCV infection as a risk factor only (Figure 3). Of note, such a relationship was not found for the number of nonfunctional *IFNL4* transcripts.

## 3. Discussion

The TCGA LIHC cohort of patients with HCC of various different etiologies was chosen as an independent source for an approach to validate previous in-house single-center gene transcription data on IFNs and ISGs in human liver tissue samples [16]. In the first place, TCGA gene expression results were found to support the concept of HCV exclusively being able to activate hepatic *IFNL4* gene transcription, also in advanced liver disease. With the exception of one patient out of 236, substantial *IFNL4* mRNA gene expression levels (>1000 FPKM-UQ) were seen only in the group of patients with HCV or HBV/HCV coinfections (seven out of 135) (Figure 2). Please note that this single patient was within the group of those with no known history of primary risk factors, meaning an absence of hepatitis serology.

TCGA, however, does not provide data on the hepatic viral load at the time of sampling for this cohort, which is a limitation of this cohort over our in-house sample. In our previous study, we were able to demonstrate a significant, positive relationship between hepatic *IFNL4* gene expression to the amount of the HCV RNA copies [16]. Such an analysis was not feasible with the TCGA sample, in the second place. The low proportion of patients with clear hepatic *IFNL4* activation within this TCGA cohort, moreover, might reflect a low proportion of patients with detectable viral loads. This might be due to measures to eradicate the virus in the long course of liver disease that precedes the development of HCC.

Different to other public data repositories, e.g., the unrestricted-access project Gene Expression Omnibus (GEO), TCGA provides controlled-access genome data derived from whole-exome sequencing data of nonmalignant tissues. This option allowed us to read-out patients’ *IFNL* genetic backgrounds and to assign patients to those who encoded a functional *IFNL4* gene and to those who encoded a nonfunctional *IFNL4* gene. Based on this assignment, in the third place, we were able to demonstrate a correlation between the number of functional *IFNL4* gene transcripts and the level of ISG expression (Figure 3). Of note, this relationship was valid for patients with HCV infection only (Figure 3).

The type I IFN gene expression was low and mostly nondetectable not only in liver samples from patients with nonviral but also from patients with HBV or HCV infections as the primary risk factors. These data on malignant tissues complement previous findings on liver tissue samples from patients with chronic HBV or HCV infection compared to patients with nonviral liver diseases or even of patients with the absence of any liver disease [10,12,13] and with findings by others [11]. The non-inducibility of hepatic type I IFNs might be advantageous for the virus and might promote a chronicity of infection. Paradoxically, it is a profound activation of ISGs that correlates with persistent infections (summarized in references [19,20]). We suppose that the hepatic activation of ISGs in liver samples from patients with chronic hepatitis C and from HCC patients with HCV infections as a risk factor, i.e., the IFN signature, is promoted by non-type I functional IFNs in the absence of type I IFNs.

Two prominent ISGs, *IRF-1* and *IRF-2*, have recently been shown to regulate the PD-L1 expression in HCC. They were suggested to be favorable for responsiveness towards immune checkpoint blockades with anti PD-L1/PD1 therapy [21,22] and to create an antitumor microenvironment in HCC [23]. Moreover, according to a meta-analysis of three large, randomized controlled phase III trials on immune checkpoint inhibitors, the responsiveness in terms of survival rates was found to be superior in patients with HCV-related HCC when compared to nonviral metabolic etiologies [9]. These data, together with mechanistic preclinical, and clinical evidence, imply an impact of diverse hepatic environments driven by etiology and characterized by PD1 expression on CD8^+^ T cells for immune surveillance and the outcome of cancer therapy [9]. Our data on an HCV-driven and *IFNL* genotype-dependent specific hepatic transcription profile are completely in line with the above descriptions, ultimately arguing for the stratification of patients with HCC according to etiology.

Taken together, after stratifying the TCGA HCC patients for etiology and after reading out their *IFNL* genotypes, gene expression data on this independent cohort of advanced disease substantiates a previous investigation on liver samples in man. The particular HCV-driven activation of *IFNL4* and the activation of ISGs that relates to host *IFNL* genotypes is seen in the malignant condition as well. The data thus add to the understanding of host immune phenotypes in HCC and underscore the importance of considering etiology as a host variate in personalized schemes.

## 4. Materials and Methods

### 4.1. TCGA Data

Analyses were based upon data generated by TCGA (phs000178.v10.p8), which was managed by the NCI and NHGRI. Specifically, the project on HCC (TCGA-LIHC; https://portal.gdc.cancer.gov/projects/TCGS-LIHC, accessed on 30 July 2018) was regarded. The access to controlled data was approved by the NIH (project ID 20041). Besides demographic (gender, age at diagnosis, and ethnicity) and clinical (tumor grade and stage) data, this study was applied on open access normalized transcriptome profiling data (FPKM-UQ, fragments per kilobase of transcript per million mapped reads upper quartile) both from the primary tumor tissue (code 01A) and corresponding nonmalignant normal liver tissue (code 11A). The following records were regarded: *IFNA2* (ENSG00000188379), *IFNA8* (ENSG00000120242), *IFNA21* (ENSG00000137080), *IFNB1* (ENSG00000171855), *IFNG* (ENSG00000111537), *IFNL1* (ENSG00000182393), *IFNL2* (ENSG00000183709), *IFNL3* (ENSG00000197110), *IFNL4* (ENSG00000272395), *Mx1* (ENSG00000157601), *IFI44* (ENSG00000137965), *IFIT1* (ENSG00000185745), *GAPDH* (ENSG00000111640), and *ALB* (ENSG00000163631).

### 4.2. Reading Out IFNL Genotypes

Reading out *IFNL* genotypes from controlled access sequencing reads was performed as described previously [24]. In brief, whole-exome sequencing (WXS) reads of nonmalignant tissue (code 11A) or peripheral blood leucocyte (code 10A) were cut down to the region spanning the *IFNL* gene cluster (chr19: 39,230,000–39,300,000) by using the BAM slice tool. Truncated sequence files were aligned to chromosome 19 of the human reference assembly GRCh38.p12 (by using NCBI genome workbench software), and the genotypes of four nucleotide polymorphisms that were in close LD to each other were read out: *IFNL3* rs4803217 (C/A), *IFNL4* rs117648444 (G/A), *IFNL4* rs12979860, and *IFNL4* rs368234815 (formerly ss469415590, now merged into rs11322783) (TT/ΔG) (Table 2).

### 4.3. Statistical Analysis

Statistical analyses were carried out using both the PC-STATISTIK software package v4.0 (Hoffmann-Software Giessen, Giessen, Germany) and RStudio v3.5.2 [25]. Nonparametric gene expression data were compared by applying the Kruskal–Wallis test or Wilcoxon test, respectively, with regard to their distribution. The correlation of gene expression data was assessed by applying Spearman’s rank regression analyses. *p*-values < 0.05 were considered statistically significant.

The exact test for the deviation from the Hardy–Weinberg equilibrium was performed using an online calculator (http://www.husdyr.kvl.dk/htm/kc/popgen/genetik/applets/kitest.htm, accessed on 20 January 2021). Linkage disequilibrium (LD) coefficients D’ and r^2^ were enquired from the WEB-based application LDlink (https://ldlink.nci.nih.gov), which refers to the data of the 1000 Genomes Project (Phase 3, Version 5).

## Figures and Tables

**Figure 1 ijms-22-07803-f001:**
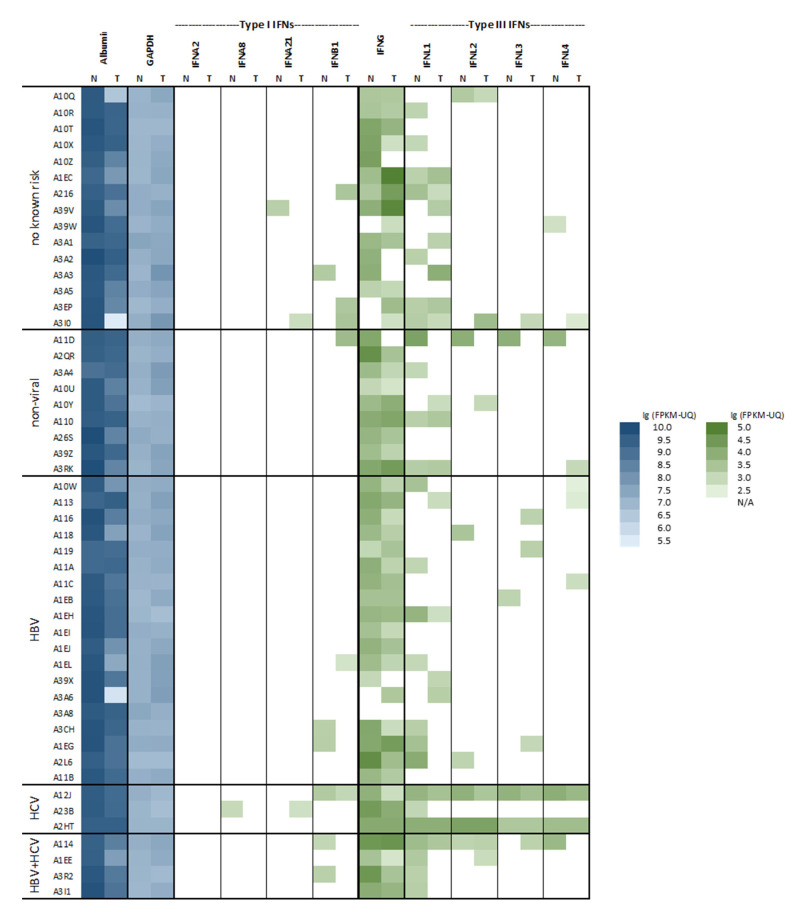
Heatmap of hepatic interferon (IFN) gene expression in nonmalignant and malignant tissues with regard to the risk factors. Transcript expression (FPKM-UQ) of an array of IFNs was plotted against HCC risk factors for 50 matched nonmalignant (N) and malignant (T) sample pairs. Expression data for two references, albumin and GAPDH, were included. Blank fields indicate the absence of any fragments (entry FPKM-UQ: 0).

**Figure 2 ijms-22-07803-f002:**
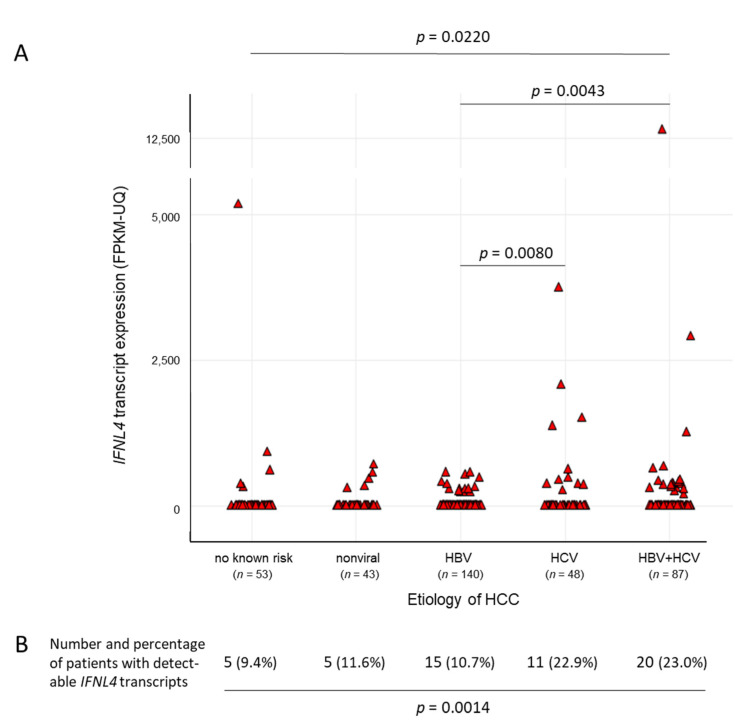
*IFNL4* transcripts in hepatocellular carcinoma (HCC) tissue samples with regard to etiology. The amount of *IFNL4* transcripts from a total of 371 TCGA human tissue specimens was analyzed with regard to HCC etiology. A comparison of patients with or without HCV infection as a risk factor for the development of HCC revealed significant differences in the distribution of *IFNL4* mRNA levels, which are given both as discrete (**A**) and categorized (**B**) data. Wilcoxon and Kruskal–Wallis tests were applied for comparisons of 2 or >2 groups, respectively (**A**), and a χ^2^-test (**B**) compared no known risk/nonviral/HBV vs. HCV/HBV + HCV.

**Figure 3 ijms-22-07803-f003:**
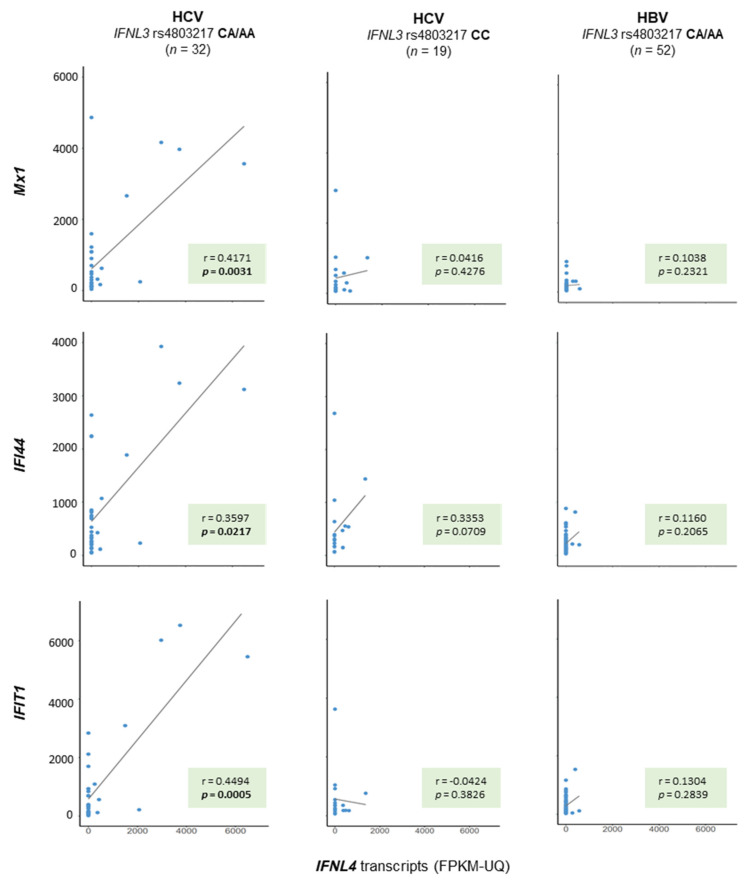
Hepatic IFN-stimulated gene (ISG) expression relates to the number of functional *IFNL4* transcripts in liver tissue specimens from patients with HCV infection as a risk factor. Patients with HCV or HBV infection as a risk factor for developing HCC were stratified with regard to those encoding for a functional IFN-λ_4_ protein (rs4803217 CA/AA) and into those who do not (rs4803217 CC), as indicated. The number of *IFNL4* transcripts was then related to the number of three ISGs—namely, *Mx1*, *IFI44*, and *IFIT4*. A Spearman rank regression analysis revealed significant positive correlations between *Mx1*, *IFI44*, and *IFIT4* and *IFNL4* transcripts in patients with an IFN-λ_4_ creative genetic background and HCV infection as a risk factor only (column 1). Such a relationship was not found for patients with HCV infection as a risk factor for HCC and an IFN-λ_4_ disruptive genetic background (column 2), nor for patients with an IFN-λ_4_ creative background but who are at risk because of an infection with HBV (column 3). The number of ISG transcripts is given in FPKM-UQ × 10^−3^.

**Table 1 ijms-22-07803-t001:** Patient characteristics.

	Total	Unknown Risk	Nonviral	HBV	HCV	HBV + HCV	*p*-Value
	(*n* = 377)	(*n* = 54)	(*n* = 43)	(*n* = 144)	(*n* = 48)	(*n* = 88)	
**Age**, mean ± SD (years)	59.5 ± 13.5	61.1 ± 17.1	67.0 ± 9.5	57.5 ± 13.0	60.2 ± 8.6	57.7 ± 14.4	<0.0001
**Gender**, m/f (*n*)	255/122	20/34	31/12	104/40	38/10	62/26	<0.0001 ^1^
							0.7385 ^2^
^3^ **Ethnicity** (*n* (%))							
White American	187 (51.0)	38 (74.5)	35 (83.3)	42 (29.6)	31 (66.0)	41 (48.2)	<0.0001
Asian American	161 (43.9)	11 (21.6)	5 (11.9)	97 (68.3)	7 (14.9)	41 (48.2)	
Black/African American	17 (4.6)	1 (2.0)	2 (4.8)	3 (2.1)	8 (17.0)	3 (3.5)	
Natives	2 (0.5)	1 (2.0)	0	0	1 (2.1)	0	
^4^ **Tumor grade** (*n* (%))							
G1	55 (14.8)	8 (15.4)	6 (14.0)	11 (7.6)	10 (21.7)	20 (23.0)	
G2	180 (48.4)	30 (57.7)	22 (51.2)	55 (38.2)	26 (56.5)	47 (54.0)	<0.0001
G3	124 (33.3)	13 (25.0)	15 (34.9)	67 (46.5)	10 (21.7)	19 (21.8)	
G4	13 (3.5)	1 (1.9)	0	11 (7.6)	0	1 (1.2)	
^4^ **AJCC tumor stage** (*n* (%))							<0.0001 ^1^
I	175 (49.6)	24 (50.0)	15 (44.1)	84 (59.6)	26 (59.1)	26 (30.2)	
II	87 (24.7)	8 (16.7)	11 (32.4)	35 (24.8)	12 (27.3)	21 (24.4)	0.2660 ^5^
III	86 (24.4)	15 (31.3)	7 (20.6)	19 (13.5)	6 (13.6)	39 (45.4)	
IV	5 (1.4)	1 (2.1)	1 (2.9)	3 (2.1)	0	0	
***IFNL3* rs4803217**							
CC:CA:AA	227:123:26	29:21:4	25:16:2	100:35:8	19:19:10	54:52:2	0.0005
MAF	0.232	0.269	0.232	0.178	0.406	0.259	0.0003

^1^ χ^2^-test, all groups included. ^2^ χ^2^-test, patients with unknown risk excluded. ^3^ Data on ethnicity available for 367/377 patients. ^4^ Data on tumor grade and stage available for 372 and 353 patients, respectively. ^5^ χ^2^-test, HBV + HCV patients excluded.

**Table 2 ijms-22-07803-t002:** *IFNL* gene polymorphism LD.

SNP	Alleles	MAF			D’ (Red) and r^2^ (Yellow)		
		AS	EUR	rs4803217	rs117648444	rs12979860	rs368234815
*IFNL3* rs4803217	C/A	0.153	0.308	1.0	0.990	0.980	0.978
*IFNL4* rs117648444	G/A	0.024	0.118	0.119	1.0	1.000	1.000
*IFNL4* rs12979860	C/T	0.156	0.309	0.944	0.120	1.0	0.997
*IFNL4* rs368234815	TT/ΔG	0.159	0.311	0.892	0.114	0.943	1.0

## Data Availability

The raw datasets supporting the reported results are available via The Cancer Genome Atlas (TCGA) web portal (https://portal.gdc.cancer.gov/projects/TCGA-LIHC).

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
