# Peer review of "Hepatic IFNL4 Gene Activation in Hepatocellular Carcinoma Patients with Regard to Etiology"

_ijms, 2021, doi:10.3390/ijms22157803_

Round 1
Reviewer 1 Report
The relationship between the type of viral infection and the genotype of an interferon-related pseudogene (IFNL4) is studied in the TCGA cohort. This study is original by levering the genotype of TCGA participants to analyze the relationship between risk-factors and gene expression signature.
The level of proof is overall fairly low. In addition to a clear association between IFNL4 genotype or expression and HCV etiology, only a mild increase in expression of some IFN-stimulated genes is shown.
For instance, it would be interesting to see if the IFNL4-high patients with CA/AA genotype have different immune infiltrates (using deconvolution tools on RNAseq such as immunedeconv or MCPcounter) than other HCV patients.
Minor comments:
Figure 1 is missing a color scale (especially, under what counts are sample shown blank)
Data not shown should be shown.
Author Response
Reply to Review Report (Reviewer 1)
The relationship between the type of viral infection and the genotype of an interferon-related pseudogene (IFNL4) is studied in the TCGA cohort. This study is original by levering the genotype of TCGA participants to analyze the relationship between risk-factors and gene expression signature.
Thank you a lot for evaluating our manuscript and for your kind estimation.
The level of proof is overall fairly low. In addition to a clear association between IFNL4 genotype or expression and HCV etiology, only a mild increase in expression of some IFN-stimulated genes is shown.
The increase of ISGs is about 2-3 fold on average depending on etiology when all patients with HCV infection as a risk factor are regarded. This data is now depicted in supplementary Figure 1S. However, by regarding the subgroup of wild-type allele carriers only, the increase is higher yet again when etiology HCV (n=29) is compared to etiology HBV (n=42) (4.3-fold for Mx1, p=.0011; 3.3-fold for IFI44, p=.0003; 3-fold for IFIT1, 0=.0300). Similar results are obtained when choosing non-viral etiology (n=18) for comparison (4.5-fold for Mx1, p=.0043; 3-fold for IFI44, p=.0114; 3.6-fold for IFIT1, p=.0460), for instance.
For instance, it would be interesting to see if the IFNL4-high patients with CA/AA genotype have different immune infiltrates (using deconvolution tools on RNAseq such as immunedeconv or MCPcounter) than other HCV patients.
Thanks a lot for this suggestion. We are highly interested in correlates to germline IFNL4 variants, as we agree that this would provide a clue towards the biological processes the expression of IFNL4 would be relevant for. We also completely agree that the composition of the immune infiltrate is an obvious trait. Similarly worthy to be analyzed to our view would be immune checkpoints, MHC-dependent antigen presentation or pattern recognition (sensing pathogen- or damage-associated molecular patterns), among others. Because of the wide array of options, we are intending to go for a separate exploratory and genome-wide comparison. This work is currently in conceptualization.
Nevertheless, prompted by your comment and supported by Tim Beißbarth and Darius Wlochowitz from the Department of Medical Bioinformatics at the UMG, we applied the R package immunedeconv on the TCGA LIHC dataset. While a first heatmap clustering and a principal component analysis failed to yield a conclusive result, univariate analyses revealed some indications/clues for associations between gene expression and cellular subtypes that appear to depend on IFNL4 genotype (!). These preliminary observations are quite interesting for us, however, they would demand further thorough evaluation and we feel that this would be beyond the scope of our manuscript.
We thus would prefer to refrain from the analysis of the single biological process ‘immune infiltrate’ at this point in favor of a more comprehensive genome-wide approach not least for reason of straightness.
Minor comments:
Figure 1 is missing a color scale (especially, under what counts are sample shown blank)
The color code has now been included. Because of the wide range of expression of overall 10 log ranks, two scales have been employed for a better resolution. Blank fields (N/A) indicate those with the entry “0” according to TCGA. This information has now been included into the legend to Figure 1. The lowest number of fragments retrieved for the sample shown was 222.
Data not shown should be shown.
The respective data that have been referred to in the last paragraph of the result section are now included as a supplementary Figure 1S.
Reviewer 2 Report
The authors present a detailed analysis of TCGA dataset and suggest a role for IFNL4 in the etiology of hepatocellular carcinoma. The manuscript is well-written and the results are presented clearly. I recommend the publication of the article once these minor suggestions are attended:
- Fig 1: Please provide the scale/colorKey for the heatmap.
- Fig 3: The authors could check if it would be good to include a larger panel of ISGs in this analysis and compare/contrast between IFN-alpha/beta induced genes vs IFN-lambda induced genes. This article gives a list of IFNL-induced genes: https://www.ncbi.nlm.nih.gov/pmc/articles/PMC3579021/
Author Response
Reply to Review Report (Reviewer 2)
The authors present a detailed analysis of TCGA dataset and suggest a role for IFNL4 in the etiology of hepatocellular carcinoma. The manuscript is well-written and the results are presented clearly. I recommend the publication of the article once these minor suggestions are attended:
Thank you a lot for evaluating our manuscript and for your kind estimation.
- Fig 1: Please provide the scale/colorKey for the heatmap.
A color key for the heatmap (Figure 1) has now been included.
- Fig 3: The authors could check if it would be good to include a larger panel of ISGs in this analysis and compare/contrast between IFN-alpha/beta induced genes vs IFN-lambda induced genes. This article gives a list of IFNL-induced genes: https://www.ncbi.nlm.nih.gov/pmc/articles/PMC3579021/
Thanks a lot for your suggestion. While type I and type III IFNs signal through specific membrane receptors, IFNAR and IFNLR, downstream signaling via the JAK/STAT pathway has been shown to culminate in the activation of the same set of transcription factors, namely ISGF3 and GAF, by both of them (Heim MH DOI: 10.1016/j.jhep.2012.10.005). Overlapping signaling pathways thus impede a discrimination between type I and type III activated genes. Beyond that, there is an overlap with type II IFNG signaling, as IFNG acts through GAF as well. Not least, both transcription factors do bind to similar DNA binding motifs, ISRE and GAS, respectively, thereby stimulating an overlapping array of genes. The prototype of an IFNG-responsive gene is CXCL10. According to the table that is referred to, CXCL10 is also among the IFNL and IFNA inducible genes, pointing out this difficulty.
The selection of the three genes we made is based on the findings of a previous approach to identify differentially induced genes in HCV infection (Patzwahl R et al doi: 10.1128/JVI.75.3.1332-1338.2001). By employing suppression subtractive hybridization (SSH), a transcriptome-wide comparison, these three genes could be demonstrated as being up-regulated in HCV-infected liver tissue specimens in man when compared to a reasonable control. Since then, they were chosen as indicators for hepatic viral infections in many instances. This information indeed was missing in our manuscript, thanks for pointing out. It now has been included into the respective (2nd) paragraph of the Introduction.